# Optimizing the Maximal Perturbation in Point Sets while Preserving the Order Type

**Luis Gerardo de la Fraga \*** and **Heriberto Cruz Hernández**

Computer Science Department, Cinvestav, Av. IPN 2508, 07360 Mexico City, Mexico;
hcruz@computacion.cs.cinvestav.mx
* Correspondence: fraga@cs.cinvestav.mx

**Abstract:** Recently a new kind of fiducial marker based on order type (OT) has been proposed. Using OT one can unequivocally identify a set of points through its triples of point orientation, and therefore, there is no need to use metric information. These proposed order type tags (OTTs) are invariant under a projective transformation which allows identification of them directly from a photograph. The magnitude of noise in the point positions that a set of points can support without changing its OT, is named the maximal perturbation (MP) value. This value represents the maximal displacement that any point in the set can have in any direction without changing the triplet's orientation in the set. A higher value of the MP makes an OTT instance more robust to perturbations in the points positions. In this paper, we address the problem of how to improve the MP value for sets of points. We optimize "by hand" the MP for all the 16 subsets of points in the set of OTs composed of six points, and we also propose a general algorithm to optimize all the sets of OTs composed of six, seven, and eight points. Finally, we show several OTTs with improved MP values, and their use in an augmented reality application.

**Keywords:** order type; fiducial markers; optimization; computer vision; augmented reality

## 1. Introduction

Fiducial tags, or fiducial markers, are used in computer vision (CV) applications for robot localization [1,2], mapping and localization of large environments [3–5], or for pose estimation in medical endoscopy [6]. Markers are also used for metric purposes, e.g., for calibration [7] and monitoring changes in distances and orientations in historic structures [8]. AprilTags [9] and Aruco [10] are perhaps the more used fiducial tags. Essentially, a fiducial marker is used in a CV application as pattern for a quick and easy detection by a computer, The position and pose of the fiducial marker with respect to a viewing digital cameras can be obtained automatically by the computer.

In this work we focus on a new kind of tags based on order type (OT) [11,12], called order type tags (OTTs), which are projective invariant and are also suitable for automatic identification and pose estimation [13]. These markers are composed of a typical black square frame with an inner white square and with black triangles inside of this last white square. Four examples of these OTTs are shown in Figure 1. The position of the triangle vertices form the set of points from which is possible to calculate its associated OT. This OT also forms a unique identifier for each OTT.

The scenario for the identification task is the following: An OTT instance is located in a scene, one image is generated through the camera sensor by targeting the camera to the scene in some position and orientation, in which the visual fiducial marker is visible in the resulting image. The obtained image is processed, i.e., the image is analyzed to identify the region in which the tag is visible to compute or to estimate the positions of the triangle vertices. These positions of the triangle vertices are taken as the elements to form a set of points, and from this set of points its OT can be calculated.

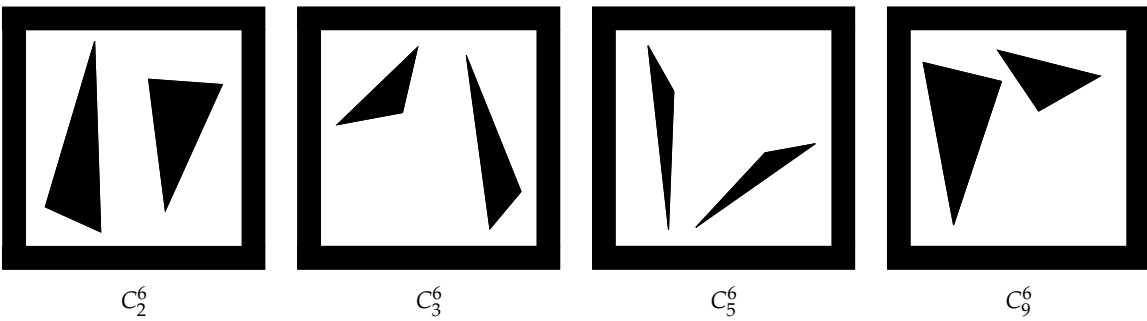

**Figure 1.** Four examples of order type tags (OTTS). The labels represent the subset number within the set $C^6$ (it is the set of all possible OTs formed with six points) in the database in [14]. More details about this notation will be given in the following sections of this work.

In [11] the authors use the point sets publicly available in [15] to build the OTTs. These data sets have already an associated maximal perturbation (MP): For a given set of points the MP is the maximal displacement (noise) that can be added to all the point positions without changing its OT. In this work we study how to optimize the MP for all the data sets in [15] with six, seven, and eight point elements. With the optimized set of points, more robust OTTs can be built.

The contribution of this work is to analyze the problem of how to maximize the MP for a set of points. We provide a general algorithm to increase the MP for a set of points. Although the problem of finding the point positions within a set that maximizes its MP appears to be an easy geometrical problem, it is not; we can only provide approximated solutions. The exact solution is still an open problem.

This paper is structured as follows: Section 2 explains the maximal perturbation definition and the algorithm to compute it. Section 3 explains the optimization of MP by hand with sets of five points. In Section 4 we present a general algorithm to optimize the MP. In Section 5 the results of this algorithm to optimize $C^6$, $C^7$, and $C^8$ will be given. In Section 6 we present a discussion, and finally in Section 7 conclusions of this work are drawn.

## 2. Order Type and Maximal Perturbation

OT is a concept from the computation geometry field, it was originally proposed by [16] and it can be seen as a way to describe sets of points based on the orientation of subsets of three points (triplets). The OT concept is valid in any dimension but in this work we will work in two dimensions, with 2D points on a plane. Formally the OT is represented with a so-called $\lambda$-matrix [16]: Each of its entries $\lambda_{i,j}$, for $i,j = \{1,2,\ldots n\}$, $i \neq j$, and $n$ points in a set, represent the number of points in the set that are on the left side of the oriented line through points $p_i p_j$, for $i \neq j$.

This number of orientations for all the triplets of points can be calculated by counting the $k$ positive double area of triangles formed by points $p_k p_i p_j$, with $k \neq i \neq j$, and $k, i, j \in \{1,2,\ldots n\}$.

In this way, the $\lambda$-matrix for the example in Figure 2 is:

$$
\begin{bmatrix}
- & 3 & 2 & 1 & 0 \\
0 & - & 1 & 3 & 2 \\
1 & 2 & - & 2 & 1 \\
2 & 0 & 1 & - & 3 \\
3 & 1 & 2 & 0 & -
\end{bmatrix}.
$$

The entries in the lower triangular $\lambda$-matrix have values $\lambda_{j,i} = n - 2 - \lambda_{i,j}$, thus only the values of the lower triangular $\lambda$-matrix, or the upper triangular $\lambda$-matrix, are independent and necessary to store a $\lambda$-matrix. Therefore only $n^2/2 - n$ memory locations are necessary to store a $\lambda$-matrix.

For two point sets *A* and *B* in a plane, where *B* is the translated, rotated and/or projected version of *A* into another plane, both sets will have the same OT. This property can be used to match the point correspondences in the two sets as in [13], and then to find the camera pose (position and orientation).

Authors in [15] analyzed all possible OTs that can be generated with sets from three up to 11 points, and those authors also provided a publicly available database [14] which contains all the possible OTs. In their database, point positions are represented with 8 bits (1 byte) up to cardinality 8. They use this binary format to store all OT instances to reduce the size of the database.

For a set of *n* points, *n*! different $\lambda$-matrices can be built, depending of how the points are labeled. Although these $\lambda$-matrices can be different, the associated OTs are not. One could select a specific $\lambda$-matrix that could help to identify the set, for example, their minimum lexicographically. In this way, the $\lambda$-matrix is one of the possibles forms to represent the OT. Another form to visualize the OT is that a $\lambda$-matrix can be built for a set of points; if the labels of the points change, the entries of the new $\lambda$-matrix will be permutations of the entries in the original built $\lambda$-matrix. Different point configurations must have different $\lambda$-matrices. All those different configurations are finite and were investigated by [15], and they are publicly available in internet [14].

There exists a single OT with three points and this forms a triangle. With four points there exists two OTs, one forms a square, and the other one forms a triangle with a point inside. With five points there exists three OTs. The set of all subsets of *n* points with different OTs will be represented as $C^n$. $C^n_i$ will represent the subset (an instance) *i* in $C^n$ with *n* points, $C^n_i \subset C^n$. We maintain the same the order for the subsets given by [15] in their database [14].

Given a set of points, its MP value is the maximal displacement that any and all the points can have safely without changing the original OT. Let *C* represent any set of points, the maximal perturbation is defined as the half of the minimal distance from any point $p_i \in C$ to any line defined by a pair of points $\overline{p_j p_k}$ in *C* with $i \neq j \neq k$. In Figure 2 the MP concept is illustrated with one example: five points are drawn in a plane, the radius of the bigger circle represents the minimal distance in any point to a line, and for this example is from point $p_3$ to line $\overline{p_1 p_2}$; the radius of the smaller circle is the MP, any point can be inside its respective smaller circle and the respective OT is kept. If points $p_2$ and $p_3$, or points $p_3$ and $p_1$, simultaneously cross the dashed line, the OT changes: The figure will be an irregular pentagon instead of a square with one point inside.

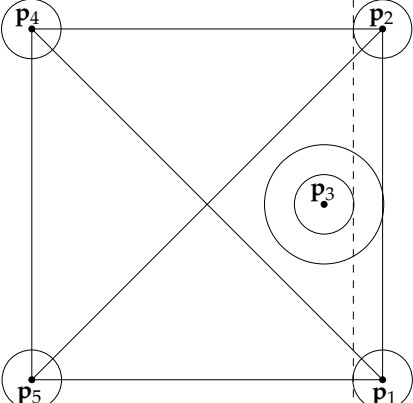

**Figure 2.** Five points in the plain. The radius of the bigger circle represents the minimal distance from any point to a line, the radius of the smaller circle is the maximal perturbation (MP). If points $p_2$ and $p_3$, or points $p_3$ and $p_1$, cross the dashed line simultaneously, the order type (OT) changes, and the figure will be an irregular pentagon instead of a square with one point inside. Points are sorted in a circular form starting with $p_1$.

In the next section, it will be optimized the MP "by hand" for every OT instance in the set $C^6$, and in the Section 4 an algorithm to optimize $C^6$ to $C^8$ will be explained.

## 3. Optimizing MP for $C^6$

We try to analyze first how to optimize the subsets of points in $C^6$ (with six points). We have a grid of $256 \times 256$ to position every point, because 8 bits were used to store every number that represents those point positions. In general, a greedy strategy is performed with the following steps:

1. Fix the points belonging to the convex hull on the grid border.
2. Position the rest of the points inside an area proportional to the number of them.
3. The points are positioned optimally. In one have four points and three of them form a triangle, that is also its convex hull, the optimal position of the fourth is in the triangle's incenter: the intersection of the three internal angle bisectors. The incenter is at equal distances from the three triangle's edges.
4. The positions are refined by a partial exhaustive search, trying to move other points not involved directly in the MP calculation.

For instance, the optimal position of point $p_3$ in Figure 2 is in the center formed by $p_1$, $p_2$, and the intersection point of line segments $\overline{p_1 p_4}$ and $\overline{p_2 p_5}$. Also, it is possible to increase the area of the referred triangle moving points $p_4$ or $p_5$; increasing the area of this triangle increases also the associated MP for the whole set of points.

The sixteen instances (or subsets) of $C^6$ are shown in Figure 3. Each one is numbered row by row (the first one in the upper corner at the left) and the number of points in their convex hull is in Table 1. The instances 1, 2, 3, 8, 9, and 10, which have 6, 5, 5, 5, 4, and 4 points in the convex hull, respectively, are trivial to optimize because of their symmetry in the convex hull points. The easiest instance is 1 because it has all its points form its convex hull, the resulted instance can be seen in Figure 4.

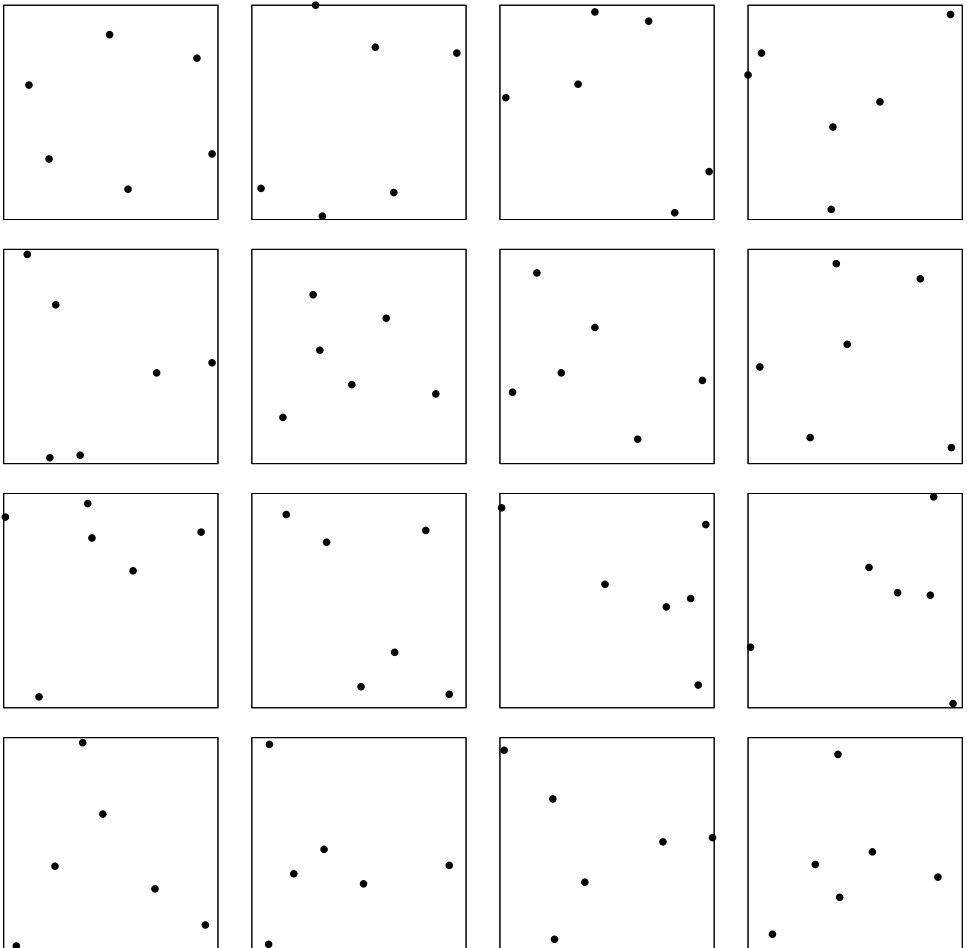

**Figure 3.** The 16 subsets of $C^6$ in the database of [14] and [15].

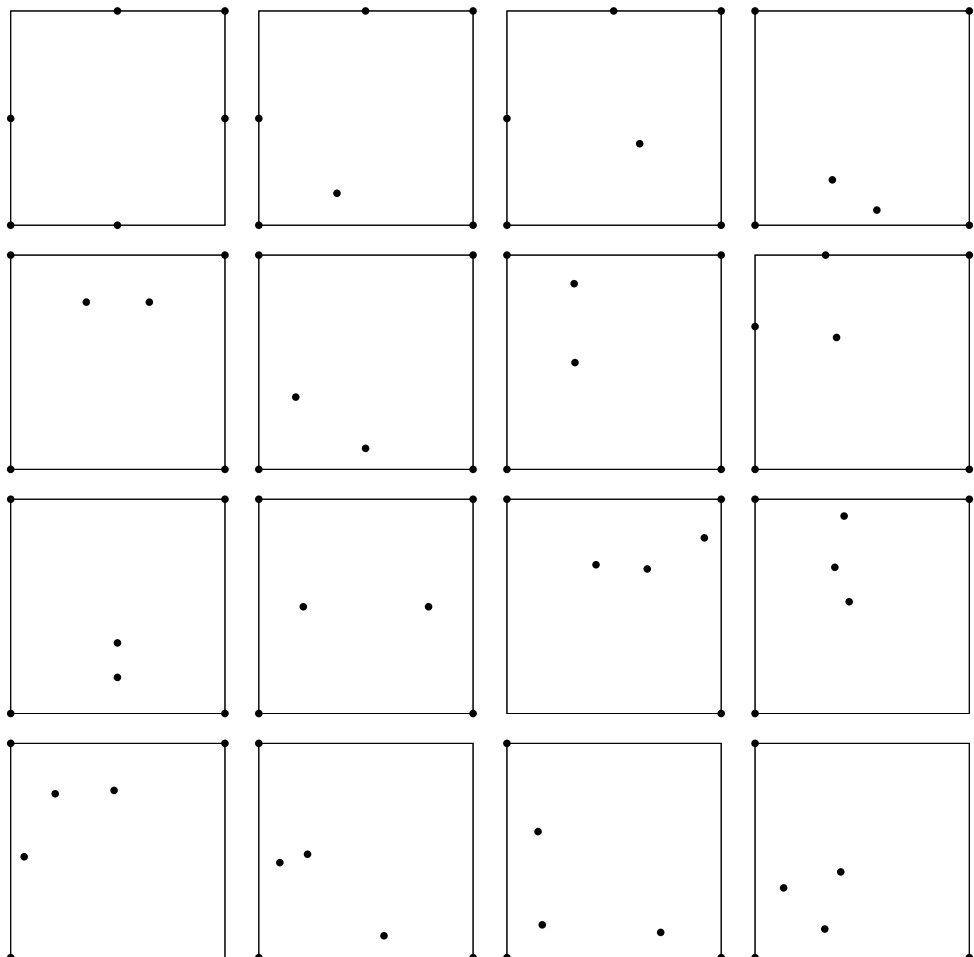

**Figure 4.** The 16 optimized subsets in $C^6$.

For the rest of the instances, each one is optimized with the aim of a different python script. The optimization procedure will be explained with one example. The instance chosen to optimize will be the fourth, it has four points in its convex hull. These four points are situated in the outer vertices of the grid, as shown in Figure 5. The point $p_3$ is positioned in the center of triangle $p_a p_b p_2$ (see Figure 5). The last point $p_4$ is positioned in the incenter of triangle $p_3 p_d p_c$. Point $p_c$ is calculated as the intersection of line segments $\overline{p_1 p_5}$ and $\overline{p_2 p_3}$. Point $p_d$ is calculated as the intersection of line segments $\overline{p_1 p_5}$ and $\overline{p_6 p_3}$. Clearly, this greedy strategy gives a small MP value (check Figure 5). Now MP can be improved if point $p_3$ is moved on the line $\overline{p_3 p_b}$. At each position on this line, the MP is recalculated at the final positions in Figure 4 is at the maximum MP value. Note that it is not the MP's global maximum. The obtained maximum MP is a local maximum. The procedure explained here is an heuristic that get good MP values—and very useful point positions—but it is not a procedure that obtains the best point positions that guarantee the MP's global maximum.

We give another example, the optimization of point subset $C_5^6$. The points $p_1$, $p_2$, $p_3$, and $p_6$ (equal to $(0,0)$, $(255,0)$, $(255,255)$, and $(0,255)$, respectively) are situated in the corners of the grid, as is shown in Figure 6 at the left. Point $p_5$ will in the incenter of triangle $p_6 p_a p_f$, where $p_a = (127,127)$ and $p_f = (127,255)$, thus $p_5 = (90,218)$. The next point $p_4$, calculated in a greedy step, will be in the center of triangle $p_5 p_e p_3$, where point e is in the intersection of lines $\overline{p_5 p_2}$ and $\overline{p_1 p_3}$, thus $p_4 = (149,196)$, with this set of points its MP value is equal to 0.63, which is very small, less than 1 unit. Instead we search for a greater MP value, with the aim of another python script, where $p_5 = (90,y)$ and $p_4 = (255 - 90, y)$, and $y$ takes the values from the point $c$ to $b$, as it is shown in Figure 6 at the right. In this way, points are symmetric with respect to the middle vertical axis, and this configuration does

not change its order type. The optimized $p_5$ and $p_4$ are $(90, 199)$ and $(165, 199)$, respectively, and the MP value for the subset is equal to 12.50.

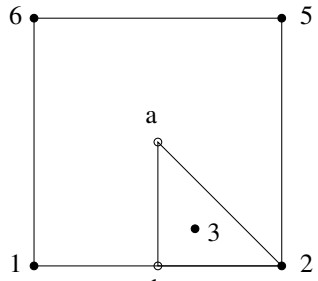 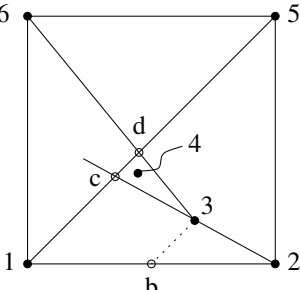

**Figure 5.** Optimizing the instance $C_4^6$. At the left the positioning of the four convex hull points and the point $p_3$. At the right the positioning of point $p_4$. Details in the text.

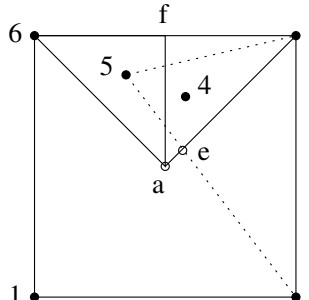 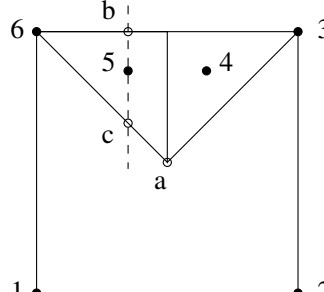

**Figure 6.** Optimizing the instance $C_5^6$. At the left the positioning of the four convex hull points, and the points $p_5$ and $p_4$. At the right is shown the dashed line where is search $p_5 = (90, y)$ and $p_4 = (255 - p_5.x, y)$. Details in the text.

A summary of the results is shown in Table 1. The higher improvements in the MP are in the easiest instances. All python scripts are publicly available (http://cs.cinvestav.mx/~fraga/OptMP.tar.gz).

We called an optimization "by hand" of these instances of $C^6$ because we were unable to generate an algorithm to calculate the initial point positions given some of the points already in the border of the grid. Here the positions are set by carefully observing the point positions in the corresponding database in [14], which are the positions that define each OT.

**Table 1.** MP values obtained by optimizing "by hand" each instance of $C^6$.

| Instance Number | Original MP | Optimized MP | # Points Convex Hull | Used for Matching |
|:---:|:---:|:---:|:---:|:---:|
| 1 | 19.44 | 28.31 | 6 | no |
| 2 | 12.27 | 19.00 | 5 | yes |
| 3 | 13.24 | 21.57 | 5 | yes |
| 4 | 8.52 | 8.75 | 4 | yes |
| 5 | 7.80 | 12.02 | 4 | yes |
| 6 | 9.36 | 12.50 | 4 | yes |
| 7 | 8.46 | 16.26 | 4 | yes |
| 8 | 13.22 | 21.21 | 5 | no |
| 9 | 7.44 | 10.53 | 4 | yes |
| 10 | 6.70 | 26.16 | 4 | no |
| 11 | 6.92 | 9.74 | 3 | yes |
| 12 | 6.59 | 7.24 | 3 | yes |
| 13 | 6.82 | 8.00 | 3 | yes |
| 14 | 8.01 | 11.69 | 3 | yes |
| 15 | 7.35 | 8.05 | 3 | no |
| 16 | 7.74 | 14.32 | 3 | no |

## 4. General Algorithm to Optimize the MP

In this section, we present our approach to automatically modify the positions of the points for a given set of points $C_1$, with a $\mathrm{MP}(C_1)$ value, to a new set of points $C_2$ with an associated $\mathrm{MP}(C_2)$ value higher or equal than the original, in such a way that the OTs of both sets are maintained unchanged, i.e., $\mathrm{MP}(C_2) \geq \mathrm{MP}(C_1)$ and $\lambda(C_2) = \lambda(C_1)$.

We were able to optimize by hand the 16 instances in $C^6$, but for $C^7$ and $C^8$ there are 135 and 3315 instances, respectively. Performing the optimization by hand for these two sets is not possible and for this reason an automatic approach is proposed.

We exploit the MP definition to iteratively modify each point position, without changing the OT. By moving each point belonging to each subset of points in such way that the OT is not changed or increased, the associated OT to that subset of points does not change. The idea is to take a point $p_i$ in the set, and try to find a new position, within its neighborhood, in such a way that the current MP is increased. If a new point position that enhances the MP is found, or at least the current MP value is not reduced, the new position is accepted. This process is repeated for all the points. The search space is discrete, and both point coordinates can have integer positions from 0 up to 255 (1 byte, as the original representation in [14]). Thus, the coordinates $(x, y)$ for a point are attempted to be moved to their eight neighborhood positions $\{(x-1, y+1), (x, y+1)(x+1, y+1)(x-1, y)(x+1, y)(x-1,\ y-1)(x,\ y-1)(x+1, y-1)\}$. We detail this idea in Algorithm 1.

The input in Algorithm 1 is an instance $l$ of $C_l^k$, that is, a set of $k$ pairs of coordinate values $(x, y)$, Then, for each point $p_i$, for $i = \{1, 2, \ldots, k\}$, each one of its eight neighborhoods $\{q\} = N(p_i)$ will be analyzed, to search for the one that increases the MP value. To perform the search, the trial set $C_{\mathrm{trial}}$ is used, which is a copy of input set $C_l^k$ with a moved point. A trial set substitutes the previous one only if the moved point increases the distance to its closest line $d_{\mathrm{trial}}$, or remains unchanged. The procedure is repeated until no more changes in the point positions can be performed, or until a maximum number of iterations is reached.

---

**Algorithm 1** Algorithm to improve $\mathrm{MP}(C_{\mathrm{in}})$ through single point displacements.

---

**Require:** Set point $C_{\mathrm{in}}$, number of MaxIterations.
**Ensure:** A set of points $C_{\mathrm{new}}$, with $\mathrm{MP}(C_{\mathrm{new}}) \geq \mathrm{MP}(C_{\mathrm{in}})$
1: $C_{\mathrm{new}} \leftarrow C_{\mathrm{in}}$ ▷ A copy of the input set of points $C_{\mathrm{in}}$
2: $i \leftarrow 0$
3: **while** $i <$ MaxIterations **do**
4: 　　$C_{\mathrm{prev}} \leftarrow C_{\mathrm{new}}$ ▷ Copy the set of points $C_{\mathrm{new}}$
5: 　　**for all** $p_i \in C_{\mathrm{new}}$ **do**
6: 　　　　Obtain the eight neighbors $\{q_k\} = N(p_i)$.
7: 　　　　Compute the distance $d_{\mathrm{current}}$ from $p_i$ to the closest line in $C_{\mathrm{new}}$
8: 　　　　Compute mp $\leftarrow \mathrm{MP}(C_{\mathrm{new}})$
9: 　　　　$p_{\mathrm{new}} \leftarrow p_i$ ▷ A new point is initialized
10: 　　　　**for** each neighbor $q \in \{q_k\}$ **do**
11: 　　　　　　$C_{\mathrm{trial}} \leftarrow C_{\mathrm{new}}$
12: 　　　　　　$C_{\mathrm{trial}} \cdot p_i \leftarrow q$ ▷ Replace $p_i$ by $q$ in $C_{\mathrm{trial}}$
13: 　　　　　　Compute the distance $d_{\mathrm{trial}}$ from $q$ to the closest line in $C_{\mathrm{trial}}$
14: 　　　　　　**if** $\mathrm{MP}(C_{\mathrm{trial}}) \geq$ mp **and** $d_{\mathrm{trial}} \geq d_{\mathrm{current}}$ **then**
15: 　　　　　　　　$d_{\mathrm{current}} \leftarrow d_{\mathrm{trial}}$
16: 　　　　　　　　mp $\leftarrow \mathrm{MP}(C_{\mathrm{trial}})$
17: 　　　　　　　　$p_{\mathrm{new}} \leftarrow q$
18: 　　　　　　**end if**
19: 　　　　**end for**
20: 　　　　$C_{\mathrm{new}} \cdot p_i \leftarrow p_{\mathrm{new}}$
21: 　　**end for**
22: 　　**if** $C_{\mathrm{prev}} = C_{\mathrm{new}}$ **then**
23: 　　　　**return** $C_{\mathrm{new}}$
24: 　　**end if**
25: **end while**
26: **return** $C_{\mathrm{new}}$

---

The complexity of Algorithm 1 is linear with respect to the number of iterations. The number of iterations can be considered unknown. The number of pixel neighbors to each pixel is constant, equal to 8. The number of points is a small number, in this study, a number in the set $\{6, 7, 8\}$. And the number of iterations is a number much bigger than the number of points.

We can not visualize a brute force approach to solve this problem. The MP value for a set of points with a fixed OT depends of the positions of the whole set. The idea of Algorithm 1 is to try to move all points and not a single one in each iteration. Also the positions of the points in the convex hull are certainly arbitrary, many of them can be tried.

## 5. MP Optimization Results for $C^6$, $C^7$, and $C^8$

Algorithm 1 was applied to all the subsets of points in $C^k$, for $k = \{6, 7, 8\}$, in [14] with a maximum number of iterations equal to 200. The results are shown in Table 2 and in Figure 7. In Table 2 the number of subsets that have a given MP value are counted, and these subsets are now represented with the set $D^n(v)$ which contains all the subsets with a MP lesser or equal to $v$. The improvement can be seen in the last row: for $D^8(9.0)$ there are now 60 optimized OTs with a MP greater than 9, when initially were only three. For $D^7(9.0)$, the number of subsets with MP greater than 9 increases from three to 33. We can observe a significant increase in the MP values: After applying our approach, OT instances that support a greater magnitude of noise are obtained. In Table 2, in the initial values on the left, we can observe the number of OT instances that support the given noise of magnitude in the first column. On the right, where we show the final results, we observe a clear improvement, i.e., most sets of points in $C^k$ increased their MP value, which increases the $D^k$ sets.

Figure 7 presents these results from another perspective, as histograms of the number of OT instances within certain ranges of MP values for the sets $C^6$, $C^7$, and $C^8$. Each subfigure shows a comparison of the initial MPs in the database in [14] and the results after applying the proposed algorithm to optimize their MP value. The four histogram comparisons clearly show an improvement of MP values. All sets of points with high MP values are increased. This case is especially notorious in Figure 7c, where we can observe that more than 2500 OT instances have their MP between 1 and 2 units. After our approach, we observe a new distribution of the bars in which the number of OTs with the same MP range is reduced to around 800, and the rest of MP ranges have a higher number of OT instances.

**Table 2.** The number of OTs for a given MP value in the original in [15] and the optimized OTs with our algorithm. The set $D$ include all the elements in $C$ which have a MP lesser or equal to the given MP value.

| MP Value | Initial MP | | | Optimized MP | | |
|---|---|---|---|---|---|---|
| | $|D^8|$ | $|D^7|$ | $|D^6|$ | $|D^8|$ | $|D^7|$ | $|D^6|$ |
| 0.5 | 3315 | 135 | 16 | 3315 | 135 | 16 |
| 1.0 | 3296 | 135 | 16 | 3301 | 135 | 16 |
| 1.5 | 1240 | 135 | 16 | 3048 | 135 | 16 |
| 2.0 | 642 | 135 | 16 | 2654 | 135 | 16 |
| 2.5 | 371 | 135 | 16 | 2240 | 135 | 16 |
| 3.0 | 231 | 86 | 16 | 1866 | 131 | 16 |
| 3.5 | 135 | 60 | 16 | 1513 | 124 | 16 |
| 4.0 | 83 | 47 | 16 | 1196 | 118 | 16 |
| 4.5 | 56 | 32 | 16 | 922 | 114 | 16 |
| 5.0 | 37 | 26 | 16 | 710 | 103 | 16 |
| 5.5 | 26 | 18 | 16 | 537 | 94 | 16 |
| 6.0 | 15 | 15 | 16 | 410 | 80 | 16 |
| 6.5 | 10 | 8 | 16 | 295 | 70 | 16 |
| 7.0 | 5 | 7 | 12 | 208 | 62 | 15 |
| 7.5 | 4 | 4 | 10 | 151 | 57 | 15 |
| 8.0 | 3 | 3 | 8 | 114 | 52 | 15 |
| 8.5 | 3 | 3 | 6 | 83 | 41 | 15 |
| 9.0 | 3 | 3 | 6 | 60 | 33 | 14 |

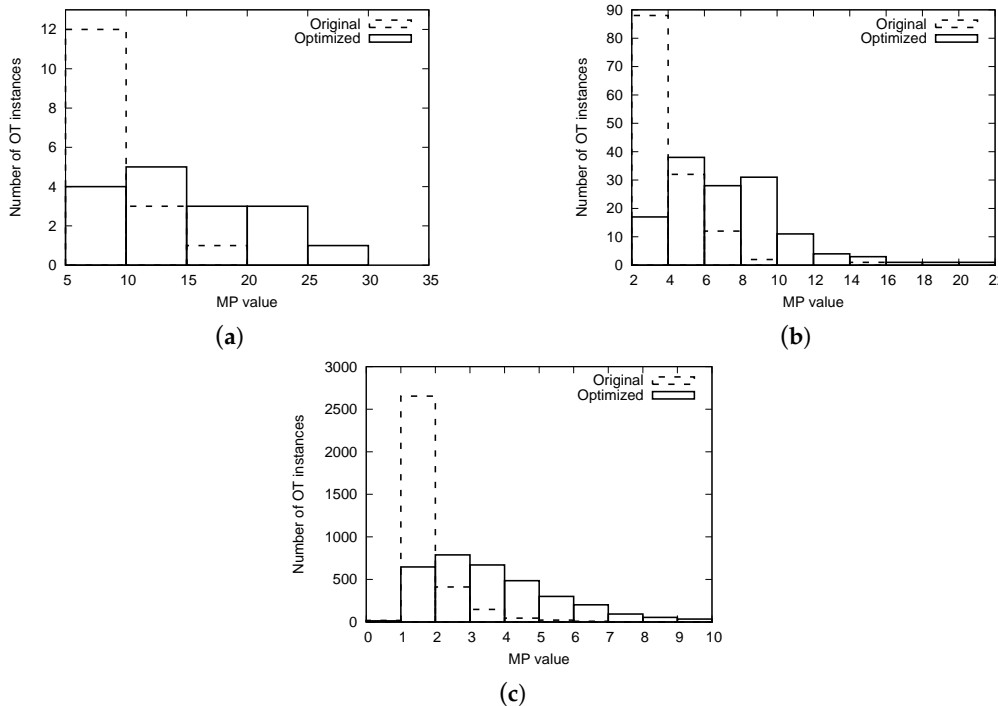

**Figure 7.** Histograms of the number of OT instances with respect to the MP values. The original MP values of the database in [15] and the ones obtained with our proposed algorithm described in Section 4 are shown. (**a**) Histogram for $C^6$; (**b**) Histogram for $C^7$; (**c**) Histogram for $C^8$.

For set $C^6$ a slight improvement was obtained with the procedure described in Section 3, in comparison with results shown in Table 2: All 16 instances have an MP value greater than 7.0 in Table 1 and in Table 2, 15 of them have a MP value greater than 7.0.

The results in Table 2 and Figure 7 clearly show that our designed Algorithm 1 automatically improves the MP for all instances in $C^6$, $C^7$, and $C^8$, and is working correctly. After applying our algorithm to the original data in [15], it improved the positions for all the subsets of points, increasing their associated MP.

Two instances of the initial and resulting sets of points are shown in each row in Figure 8. In this figure, the left column shows the initial set of points and their associated MP value, the column at the right shows the obtained results in the last iteration of our approach, and their associated MP value.

In Figure 9 we can compare the OTTs made with the $C_5^6$ and $C_9^6$ subsets before and after the optimization by hand, as explained in Section 3. These two markers without their optimized point positions were shown previously in Figure 1. The markers are detected first in the input image as the black object with four corners; then, for the markers without optimize (in the left column of Figure 9), the black triangles inside the white zone are extracted. For the optimized markers built with $C_5^6$ and $C_9^6$, these set of points have a convex hull that is also a square, thus the white zone was eliminated and white triangles are used instead. The elimination of this white zone (see the markers in the second column of Figure 9) increases the area cover by the optimized $C_5^6$ and $C_9^6$ set of points. A bigger area makes these optimized markers easier to recognize at farther distances.

Finally, an augmented reality application that use these two optimized OTTs is shown in Figure 10: Above each marker is drawn a virtual object. The point positions are calculated as the vertices of the black square and the vertices of the two white triangles. The extraction is made as follows: (1) A binarization is applied to the input image, (2) all the black components are labeled. For each black component in the image: (3) the perimeter is calculated, (4) the position of the first pixel in the perimeter is a vertex, (5) the position of the farthest pixel in the perimeter to the first vertex found, is the second vertex, and (6) the other two vertices are calculated as the positions of the two farthest pixels to the line that joins the two first vertices found. A similar process is made to calculate the

three triangle vertices. These rough vertices positions are enough to calculate the match with the point positions of the marker model using the $\lambda$-matrices. To calculate the marker pose we use homography, and then it is necessary to refine the vertices positions to a subpixel precision by extracting the points among each pair of vertices, fitting a line with those points, and calculating the intersection point for each two lines. In Figure 10, each virtual object can be moved interactively when the respective OTT is also moved. This application uses the Qt and OpenGL libraries.

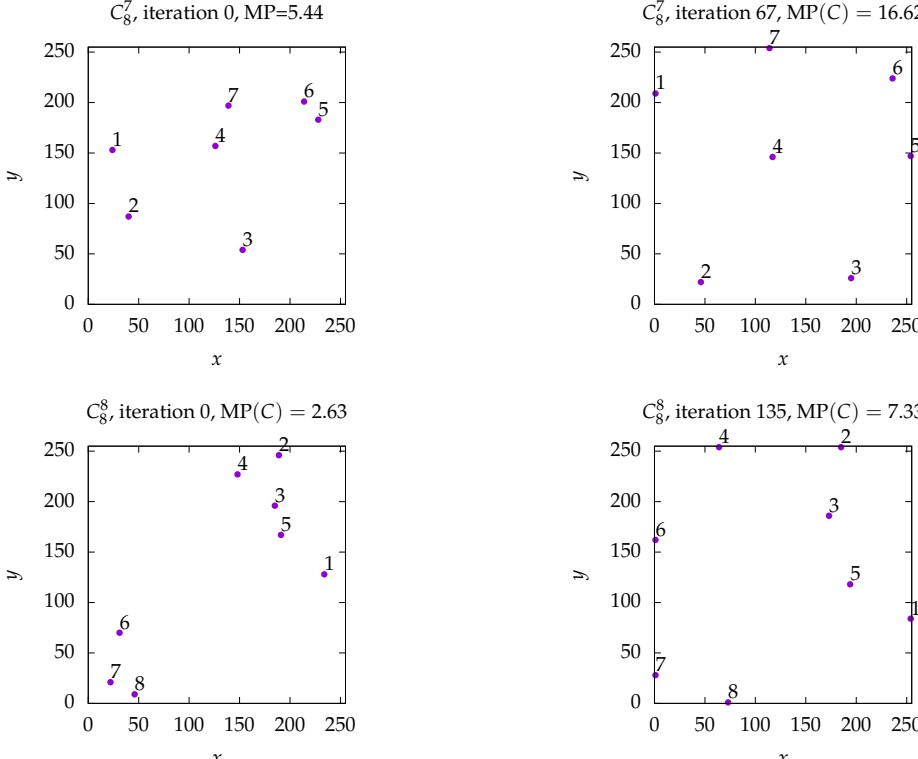

**Figure 8.** Two OT instances with improved MP value, one per row. To the left the initial set of points and to the right the set of points after applying Algorithm 1. Note how the set of points tend to cover most of the available space.

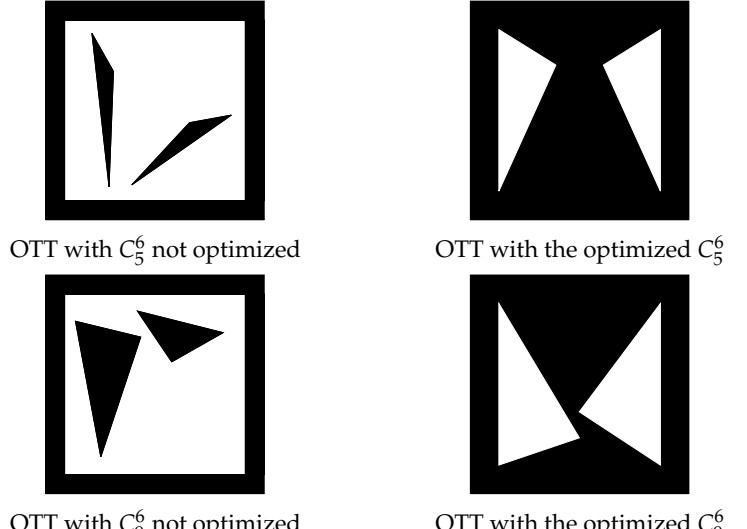

**Figure 9.** Two examples of order type tags (OTTs) built with the subsets of points $C_5^6$ and $C_9^6$ without the original point positions in the database in [14], and with the corresponding optimized point positions by hand.

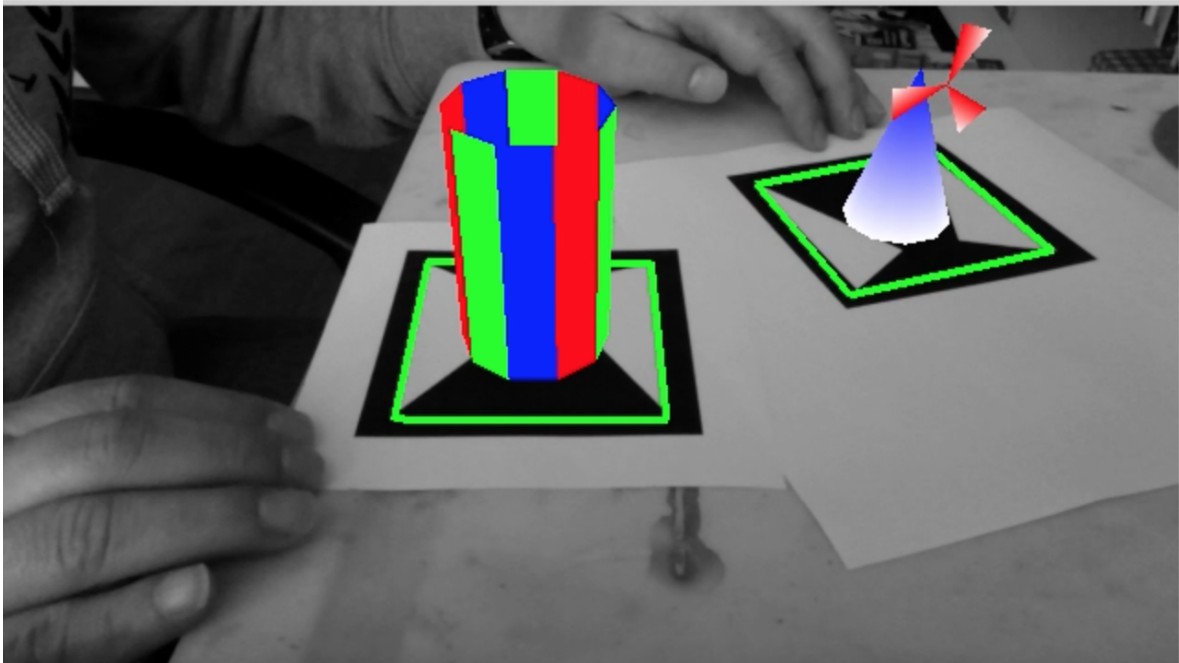

**Figure 10.** A picture of an augmented reality application that shows two virtual objects drawn above the optimized markers $C_5^6$ and $C_9^6$. These markers are also shown at the right of Figure 9.

## 6. Discussion

According to the results presented in Sections 3 and 4, we were able to improve the MP value for most of the processed OT instances. The improvement is very noticeable in histograms in Figure 7. In the comparison of histograms, we increased the number of OT instances with higher MP values, and also we reduced the number of OT instances with small MP values.

The MP value is only defined by the distance of the point to the closest line (defined by each pair of points) in the entire set of points. Although we are only analyzing one point and its distance to the closest line, in turn, this distance will define the current MP for the whole set of points. This is because the MP depends on only three points from the whole set of points. We could call these three points the critical points. In this sense, we could think that there is no reason for caring about displacing critical points that do not directly define the OT. Nevertheless, in some situations, the displacements of the non-critical points is necessary to allow future possible displacements for the critical points. For instance, consider a case with four points, these form an equilateral triangle and the fifth located at the center of the triangle. In that OT instance, the $MP(C)$ depends on the distance of the central point to any of the three lines of the triangle. In a case like that, we see that we can not perform any movement on the central point without reducing the MP value. In order to make possible a movement in the central point that improves the MP value of the set, we first should move some of the points that form the triangle. With a movement of one or more of those points that define the external triangle we could make some space in such a manner that, in a subsequent search of a new location for the central point, we can find a new location that improves the MP for the set. This is exactly how our approach works. The algorithm tries to increase the distance of every point to its closest line, to try to expand the set of points in such a way that the critical points could be moved to new positions that improve the MP.

Optimizing by hand instances with more than six points is difficult. However, we can position the convex hull points on the marker corners. It could be very interesting to optimize by hand the instances in $C^7$ as those have four points in the same maker corners, and the other three points form a single triangle, and so its detection process will be simpler. We left this idea as a future work. It could be possible to fix the four points in the convex hull to the grid corners, and optimize the rest of the

points with our proposed algorithm. In such a case, it will be necessary to carefully analyze by hand the initial point positions, which also define the associated order type for the set of points.

Also left as future work, is the analysis for an algorithm to calculate or to search the global optimum MP for a given set of $n$ points. To find the point positions that correspond to the global optimum MP is still an open problem.

## 7. Conclusions

We analyze how MP can be optimized in the set $C^6$, with six points. We also propose a general algorithm to improve the MP for $C^7$, $C^8$, and $C^9$. The proposed algorithm performs one single distance point displacement at each iteration, based on the definition of the MP, which states that we can move any point in a set $C$ to a new location without changing its OT if this movement is smaller than $MP(C)$. We showed that our approach improves most of the OT instances of the database provided by [15]. The proposed approach improves the set of points in terms of the MP value, and thus it improves the performance for the markers based on OT, as these can resist more noise when their point positions are detected in an image. We show the use of these OT markers in an augmented reality application.

**Author Contributions:** L.G.d.l.F. and H.C.H. conceived, designed, and performed the experiments; L.G.d.l.F. and H.C.H. analyzed the data; L.G.d.l.F. wrote the paper.

**Funding:** This research received no external funding.

**Acknowledgments:** The second author acknowledges support from the Conacyt to pursue his Ph.D. studies.

**Conflicts of Interest:** The authors declare no conflict of interest.

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
