# Peer review of "Optimizing the Maximal Perturbation in Point Sets while Preserving the Order Type"

_mca, doi:10.3390/mca24040097_

Round 1

Reviewer 1 Report

Order Type Tags (OTTs) based on OT are used in Computer Vision applications such as robot localization and localization of large environments. A higher value of the Maximal Perturbation (MP) makes an OTT instance more robust to perturbations in the points positions. This paper propose a algorithm optimize the MP in Point Sets preserving its OT that can modify automatically the positions of the points. The experimental results prove the validity of the algorithm, agree to accept but need to make some modifications.Here are my opinions.

Section 2 describes definition of OT too much and introduce less about MP content. In Section 2, The definition Cof is not very clear. In Section 3, the contents of example in the third paragraph are unclear, and the sentences are not grammatical. Table 1 shows a summary of the all obtained results, and it can be placed in the end of this section(after figure 9). In algorithm 1, pnew is not defined.  Section 4 doesn't introduce innovations about your approach.Hope to improve. In Section 5, you don't explain what the reason for the experimental results are getting better.

Author Response

Reviewer 1 (R1):

Order Type Tags (OTTs) based on OT are used in Computer Vision applications such as robot localization and localization of large environments. A higher value of the Maximal Perturbation (MP) makes an OTT instance more robust to perturbations in the points positions. This paper propose a algorithm optimize the MP in Point Sets preserving its OT that can modify automatically the positions of the points. The experimental results prove the validity of the algorithm, agree to accept but need to make some modifications.Here are my opinions.

Observation 1:

  R1: Section 2 describes definition of OT too much and introduce less about MP content.

  Answer:

  We appreciate all your comments that help us to improve this work.

    The title of Section 2 is changed to:

    2. Order Type and Maximal Perturbation

Observation 2:

  R1: In Section 2, The definition Cn of is not very clear.

  Answer: The text:

  "We are going to represent as Cn the set which contains all the set of points with cardinality n with different OT."

  It is changed to:

  The set of all subsets of $n$ points with different OT will be represented as $C^n$.

Observation 3:

  R1: In Section 3, the contents of example in the third paragraph are unclear, and the sentences are not grammatical.

  Thank you for your observation. The sentences has been corrected.

Observation 4:

  R1: Table 1 shows a summary of the all obtained results, and it can be placed in the end of this section(after figure 9).

  This Table 1 now has been placed in the end of Section 3.

Observation 5:

   In algorithm 1, pnew is not defined.

   Answer: It is a point and is initialized in line 9 of the Algorithm 1. It is also new point.

      A comment is added to the algorithm to clarify this observation.

Observation 6:

  Section 4 doesn't introduce innovations about your approach.

  Answer: The next paragraph is added after the first paragraph in Section 4:

    We were able to optimize by hand the 16 instances in $C^6$, but for $C^7$ and $C^8$ there are 135 and 3315 instances, respectively. Performing the optimization by hand for these two sets is not possible and for this reason an automatic approach is proposed.

Observation 7:

  Hope to improve.

  In Section 5, you don't explain what the reason for the experimental results are getting better.

  Answer: Thank you for your observation. This paragraph has been added in Section 5:

  Results in Table 2 and Figure 7 shows clearly that our designed Algorithm 1 to improve automatically the MP for all the instances in $C^6$, $C^7$, and $C^8$ is working correctly. After to apply our algorithm to the original data in [11], it improved the positions for all the subsets of points, increasing their associated MP.

Reviewer 2 Report

The paper presents some results, yet there is no accurate analysis of convergence of the algorithms nor theoretical guarantees. Many results are obtained through optimization by hand. 

A more formal treatment would have been preferred also considering the journal. It is suggested at least to explain the complexity of the problem and to compare with brute force search, explaining why the latter is unfeasible.

Other remarks:

-Line 18. Markes are also used for metric purposes, e.g. for calibration and monitoring changes in distances and orientations (see e.g. 10.3390/electronics7120421 and 10.3390/jimaging4080099)

-Line 18: Please, cite Aruco and Aruco diamonds besides April Tags since they are also very well know. I suggest to add this citation: ""Speeded up detection of squared fiducial markers", Francisco J.Romero-Ramirez, Rafael Muñoz-Salinas, Rafael Medina-Carnicer, Image and Vision Computing, vol 76, pages 38-47, year 2018"

-Lines 20-12: "From the fiducial marker, it is obtained automatically by the computer its position and pose with respect to a viewing digital camera" Rephrase somehow for improved clarity. For instance "The position and pose of the fiducial marker with respect to a viewing digital cameras can be obtained automatically by the computer".

-Line 43-44: "in Section 2 the maximal perturbation definition and thealgorithm to compute it is explained"-> "Section 2 explains the maximal perturbation definition and the computational algorithm".-Line 50: Please, for improved clarity,  add somewhere that you talking about points in the plane.

-Line 56: Please, correct the formula in Lambda_[j,i] = n - 2- Lambda[i,j]

-LIne 66: the actual number of labelling should be n!, that is the number of permutation of n elements.

-Line 74: "With 3 points exists a single OT and this forms a triangle"->"There exists a single OT with 3 points and this forms a triangle".

-Line 134: "A summary of the all obtained results is shown in Table 1."->"A summary of the results is shown in Table 1."

-Figure 9: Please, explain why black and white colors are invered

-Figure 10: describe how you got this results. How accurate is the estimation of a) the centroid of the square positioning (that dependes on the boundary square alone?) and of b) the orientation of the marker? 

Author Response

Reviewer 2 (R2):

  The paper presents some results, yet there is no accurate analysis of convergence of the algorithms nor theoretical guarantees.

  Many results are obtained through optimization by hand.

  A more formal treatment would have been preferred also considering the journal. It is suggested at least to explain the complexity of the problem and to compare with brute force search, explaining why the latter is unfeasible.

  Answer 1: At the end of Section 4, the following two paragraphs are added:

    The complexity of Algorithm 1 is linear with respect to the number of iterations. The number of iterations can be considered unknown. The number of pixel neighbors to each pixel is constant, equal to 8. The number of points is a small number, in this study is a number in the set $\{6,7,8\}$. And the number of iterations is a number much bigger than the number of points.

    We can not visualize a brute force approach to solve this problem. The MP value for a set of points with a fixed OT depends of the positions of the whole set. The idea of Algorithm 1 is to try to move all points  and not a single one in each iteration. Also the positions of the points in the convex hull are certainly arbitrary, many of them can be tried.

R2: Other remarks:

-Line 18. Markes are also used for metric purposes, e.g. for calibration and monitoring changes in distances and orientations (see e.g. 10.3390/electronics7120421 and 10.3390/jimaging4080099)

-Line 18: Please, cite Aruco and Aruco diamonds besides April Tags since they are also very well know. I suggest to add this citation: ""Speeded up detection of squared fiducial markers", Francisco J.Romero-Ramirez, Rafael Muñoz-Salinas, Rafael Medina-Carnicer, Image and Vision Computing, vol 76, pages 38-47, year 2018"

-Lines 20-12: "From the fiducial marker, it is obtained automatically by the computer its position and pose with respect to a viewing digital camera" Rephrase somehow for improved clarity. For instance "The position and pose of the fiducial marker with respect to a viewing digital cameras can be obtained automatically by the computer".

-Line 43-44: "in Section 2 the maximal perturbation definition and the algorithm to compute it is explained"-> "Section 2 explains the maximal perturbation definition and the computational algorithm".-Line 50: Please, for improved clarity,  add somewhere that you talking about points in the plane.

-Line 56: Please, correct the formula in Lambda_[j,i] = n - 2- Lambda[i,j]

-LIne 66: the actual number of labelling should be n!, that is the number of permutation of n elements.

-Line 74: "With 3 points exists a single OT and this forms a triangle"->"There exists a single OT with 3 points and this forms a triangle".

-Line 134: "A summary of the all obtained results is shown in Table 1."->"A summary of the results is shown in Table 1."

Answer: We appreciate these observations. Thank you very much.

  All these changes have been made in the document.

R2: -Figure 9: Please, explain why black and white colors are inverted

Answer. In lines 190-192 :

190                                         As we fix the points in the convex hull on the corner’s

191 grid, we are not using the white inner square for the optimized OTTs. Also, as the size zone for

192 the triangles is increased in the optimized OTTs, these markers can be recognized easily.

This text is changed to:

The markers are detected first in the input image as the black object with four corners; then, for the markers without optimize (in the left column of Figure 9), the black triangles inside the white zone are extracted. For the optimized markers built with $C_5^6$ and $C_9^6$, these set of points have a convex hull that is also a square, thus the white zone was eliminated and white triangles are used instead. The elimination of this white zone (see the markers in the second column of Figure 9) increases the area cover by the optimized $C_5^6$ and $C_9^6$ set of points. A bigger area makes these optimized markers easier to recognize at farther distances.

R2. -Figure 10: describe how you got this results. How accurate is the estimation of

  a) the centroid of the square positioning (that depends on the boundary square alone?)

  and of b) the orientation of the marker?

 Answer: In lines 192-195:

192                                                                                    Finally, an

193 Augmented Reality application that use these two optimized OTTs is shown in Figure 10: above each

194 marker is drawn a virtual object. Each virtual object can be moved interactively when the respective

195 OTT is also moved.

These phrases are changed to:

    Finally, an Augmented Reality application that use these two optimized OTTs is shown in Figure 10: above each marker is drawn a virtual object. The point positions are calculated as the vertices of the black square and the vertices of the two white triangles. The extraction is made as follows: 1) A binarization is applied to the input image, 2) All the black components are labeled. For each black component in the image: 3) Its perimeter is calculated, 4) The position of the first pixel in the perimeter is a vertex; 5) The position of the farthest pixel in the perimeter to the first vertex found, is the second vertex; 6) The other two vertices are calculated as the positions of the two farthest pixels to the line that joint the two first vertices found. A similar process is made to calculate the three triangle vertices.

These rough vertices positions are enough to calculate the match with the point positions of the marker model using the $\lambda$-matrices. To calculate the marker pose we use the homography, and then it is necessary to refine the vertices positions to a subpixel precision by extracting the points among each pair of vertices, fitting a line with those points, and calculated the intersection point for each two lines. In Figure 10, each virtual object can be moved interactively when the respective OTT is also moved. This application uses the Qt and OpenGL libraries.

Round 2

Reviewer 2 Report

Authors have properly replied to the comments made. The paper might now be considered for publication.